# From Young to Older, the 4 Phases Method Is Efficient in Promoting Quick Weight, BMI, and Waist Circumference Reductions

**DOI:** 10.3390/healthcare10081398

**Published:** 2022-07-27

**Authors:** Edson Ramuth, Sylvia Ramuth, Tamaris R. R. Pavão, Kimberlly B. Biacchi, Andre L. L. Bachi

**Affiliations:** 1Faculty of Medical Sciences of Santa Casa de São Paulo, São Paulo 01224-001, Brazil; edsonramuth@uol.com.br; 2São Camilo University Center, São Paulo 04263-200, Brazil; 3Biomedical Science, Method Faculty of São Paulo (FAMESP), São Paulo 04046-200, Brazil; tataroseirapav@gmail.com; 4Biomedical Science, Franciscana University, Rio Grande do Sul 97010-030, Brazil; kimberllybiacchi@yahoo.com.br; 5Post-Graduation Program in Health Sciences, Santo Amaro University (UNISA), São Paulo 04743-030, Brazil; allbachi77@gmail.com

**Keywords:** obesity, overweight, weight loss, body mass index, waist circumference, low-carbohydrate diet, very low carbohydrate diet, ketogenic diet, physical activity

## Abstract

Background: Investigate the effectiveness of the scientific 4 Phases Method, a methodology developed by EMAGRECENTRO, which is based on a ketogenic approach (total carbohydrate intake <40 g/day; including fibers) associated with health coach assistance, in promoting reductions in body weight, body mass index (BMI), and waist circumference after 5 weeks of methodology application. Methods: Record files from 354 individuals, both sexes, aged between 18 and 67, who took part in the 4 Phases Method were used to develop this study. Age, sex, weight, height, BMI, waist circumference measurement, and the presence of ketone bodies in the urine were evaluated before (baseline) and after 5 weeks of the 4 Phases application. Results: In general, a significant reduction in body weight (−7.8 ± 1.2 kg, *p* < 0.0001), BMI (−2.8 ± 0.4 kg/m^2^, *p* < 0.0001), and waist circumference measurement (−7.6 ± 0.4 cm, *p* < 0.0001) was found after the application of the 4 Phases Method, regardless of age, gender, and BMI. Conclusions: Taken together, our results demonstrated that the 4 Phases Method was able to promote significant body weight, BMI, and waist circumference reductions in the short term, particularly by associating a ketogenic intake strategy with a regular close follow-up weekly consultation with a health coach assistance.

## 1. Introduction

According to the World Health Organization (WHO), between 1975 and 2016, the number of people with obesity in the world almost tripled, which means approximately 13% of the world’s adult population (11% of men and 15% of women) were people with obesity in 2016. This fact leads the WHO to consider obesity as the global epidemic of the 21st century [1], even above the current COVID-19 pandemic [2]. Particularly, people with overweight and obesity are closely associated with the development of hypertension, cardiovascular disease, diabetes, dyslipidemia, and some types of cancer, among other diseases and comorbidities [3].

Based on the WHO classification, individuals with a body mass index (BMI = kg/m^2^) lower than 18.5 are considered as underweight, between 18.5 and 24.9 are considered as normal weight, between 25 and 29.9 are considered people with overweight, and equal and higher than 30 are considered people with obesity [1]. Beyond BMI, the WHO also admits the assessment of the occurrence of abdominal obesity by evaluating the waist circumference. Some waist circumference cutoffs have been related to BMI categories, showing that circumference thresholds of more than 90, 105, and 115 cm for women, and more than 100, 110, and 125 cm for men are closely associated with an increased health risk with BMI between 25 and 29.9 kg/m^2^, between 30 and 34.9 kg/m^2^, and higher than 35 kg/m^2^, respectively [4]. Furthermore, other associations between waist circumferences and anthropometric measures, in which the values used to classify abdominal obesity are more than 88 cm for women and more than 102 cm for men, are of high relevance for the risk of developing many diseases, comorbidities, or even mortality [5]. In fact, obtaining simple measurements, such as height, weight, and circumference, especially of the waist, in addition to being easy, of low cost, and applicable for all age groups, and can provide relevant data regarding the health status of people [6].

It is broadly known that individuals with overweight and obesity are two closely interchangeable terms, as these conditions are characterized by excessive accumulation of fat in adipose tissue [1,7]. Concerning the literature, the main causal factor for people with overweight is an energy imbalance between calories consumed and calories expended. Particularly, people with overweight have been shown to be attracted to high-carbohydrate, low-fat diets [8,9], while individuals with obesity generally also show a preference for highly processed foods, especially carbohydrates or refined sugars, rather than less processed foods [10,11]. Thus, even with a lower-fat diet, increased carbohydrate intake undoubtedly leads to the development of overweight and obesity [7,12].

Beyond these factors, the increase in obesity prevalence is more strongly related to lower physical activity levels. There are pieces of evidence that low levels of physical activity can increase weight gain. Furthermore, it is reported in the literature that there is a close association between the sedentary lifestyle, which reduces energy expenditure, and overconsumption of food with high energy that leads individuals to develop overweight and obesity [13,14].

To try to reverse this scenario, a prominent strategy that involves very low carbohydrate intake has grown in popularity in recent years, due to its capacity to promote weight loss. Among some strategies, the ketogenic diet (KD) stands out, which advocates the use of ketone bodies to guarantee the body’s energy supply rather than carbohydrates. Therefore, the reduction in carbohydrates intake drives the body to break down fat (mainly fatty acids), which promotes ketone body formation by the liver, leading to weight loss in response to the consumption of fat, especially in the abdominal region [15].

Based on the remarkable capacity of the ketogenic intake strategy to promote weight loss, EMAGRECENTRO, a specialized weight loss and aesthetic spa, developed the 4 Phases Method, an exclusive methodology proposed to achieve weight loss not only by a ketogenic approach food orientation but also by including regular health coach assistance. Corroborating this methodology, a recent systematic review and meta-analysis [16] reported that very low carbohydrate intake resulted in significant weight loss in the short term [−7.48 kg (95% CI −9.63 to −5.34; I^2^ = 0%), after 1 month of very low carbohydrate intake diet]; in the intermediate term [−16.76 kg (95% CI −19.08 to −14.43; I^2^ = 25%), after 4 to 6 months with very low carbohydrate intake diet]; and in the long term [−21.48 kg (95% CI −28.40 to −14.56; I^2^ = 0%), after 12 months with very low carbohydrate intake diet]. There were improvements in body composition parameters when compared to other weight loss interventions of the same duration, as well as a great effect in reducing BMI [short-time = −3.25 (95% CI −3.86 to −2.63; I^2^ = 0%; intermediate time = −6.16 (95% CI −7.04 to −5.28; I^2^ = 74%; and long-time = −7.11 (95% CI −8.84 to −5.38; I^2^ = 0%)], fat mass [in general of −11.12 kg; 95% CI −14.26 to −7.97; I^2^ = 80%), and waist circumference (in general of –16.53 cm; 95% CI –19.71 to –13.36; I^2^ = 69%). Furthermore, the authors also highlighted that in this intervention, the presence of a health professional, as a health coach, is a safe approach to avoid common side effects that are clinically mild and easy to control, and recovery is often spontaneous [16]. The health coach is also a fundamental piece of behavior modification that seems more effective in treating obesity than any traditional method, but it has not been widely applied [17].

It is of utmost importance to mention that EMAGRECENTRO, a recognized center in this field, holds all copyrights for the 4 Phases Method, and proposes a methodology that promotes a natural weight loss, as it does not use medications and/or surgical interventions, and therefore it can be performed by anyone who aims to lose weight.

Based on these pieces of information, in this study, we aimed to evaluate the effectiveness and safety of the 4 Phases Method in promoting weight, BMI, and waist circumference reductions after 5 weeks of application of this methodology in participants of different ages, gender, and BMI.

## 2. Material and Methods

### 2.1. Subjects and Study Design of the 4 Phases Method

The present retrospective study was developed by using data obtained from the follow-up evaluation forms of 354 customers of EMAGRECENTRO, a clinic specialized in weight loss and aesthetic procedures. All data used in this study (*n* = 354) were collected between January 2015 and December 2020, from the medical records of individuals who took part in the 4 Phases Method, an exclusive weight loss program developed by EMAGRECENTRO.

To perform this study, firstly we obtained the approval of the Ethics Research Committee of Santo Amaro University (number: 4928459). After that, all the EMAGRACENTRO franchises were contacted not only to explain the study context and objectives but also to ask for help to obtain the data necessary for the study’s development. We received data from 119 franchises, located in different states of Brazil, initially analyzed according to the inclusion and exclusion criteria described below. After this analysis, we again contacted the EMAGRACENTRO franchises for written informed consent from individuals who wanted to participate in this study, thus allowing us to use their medical records. While informed consent was received directly by some participants, where individuals were not contactable by the EMAGRECENTRO franchises (due to changed address, e-mail, and cellphone/telephone number), we obtained written informed consent for use of those data directly from the EMAGRECENTRO franchises through a specific document, which not only was previously approved by the Ethics Research Committee of Santo Amaro University but also is a legal and mandatory method in Brazil. Based on these pieces of information, data from 354 individuals were included and used to perform the present study.

All data used in this study were obtained from individuals between 18 and 70 years old (inclusion criteria 1), who presented, in their medical records, all data necessary for the development of this study (inclusion criteria 2), who have not undergone any other weight loss procedure/treatment (inclusion criteria 3), and who agreed to participate in this study, through personal consent or, in the impossibility of obtaining this, through the consent of the franchisee by a specific document, previously approved by the Ethics Committee (inclusion criteria 4). The exclusion criteria were: (a) EMAGRECENTRO franchises that opened after 2020; (b) or that do not have weight loss data for at least 10 weeks by 2020; (c) individuals who have undergone treatment other than the 4 Phases Method; and (d) individuals who do not present in their medical records the data necessary for the development of this study.

Anthropometric data [weight (kg), height (m), body mass index (BMI, kg/m^2^), and waist circumferences (cm)] from women and men, aged between 18 and 70, were assessed on two different occasions: at the beginning and 5 weeks after submission to the 4 Phases Methodology.

The 4 Phases Method is applied according to the following protocol:

Firstly, for individuals that need to lose weight, due to medical advice, an evaluation in person at EMAGRECENTRO to verify the need for weight loss, and, consequently, whether the 4 Phases Method should be applied is recommended.

After that, since these individuals are eligible for the 4 Phases Method, a global assessment is performed to test for the presence of comorbidities, use of continuous medications, and presence/absence of allergies, as well as to obtain data on their personal and family history. In addition, anthropometric measurements are also evaluated, such as weight and height using a scale and a stadiometer, both digital, and waist circumference is also assessed using a tape measure positioned at the umbilicus level and after breathing out. These data were obtained from barefoot individuals wearing swimwear. After the evaluation, individuals presenting a BMI higher than 25 kg/cm^2^ and an increased waist circumference (>88 cm for women and >102 cm for men [5]) are introduced to the 4 Phase Method and weight loss goal is determined for the first 5 weeks—an average of approximately 5 to 10% of total weight.

In Phase 1, detoxication occurs due to reducing total carbohydrate intake to 40 g/day (including fibers) for 1 week. After this week, Phase 2 will begin, and rapid weight loss lasting 4 weeks is expected (Figure 1). After reaching the weight loss goal, individuals will then follow Phase 3 for a duration of 4 weeks, where food groups with higher amounts of carbohydrates are reintroduced, and then Phase 4, where the focus is on nutritional education and weight maintenance for 16 weeks, in order for individuals to learn how to eat and how to not regain the weight that has already been lost.

However, it is of utmost importance to point out that individuals have free access to ingest proteins, fats, salads, and vegetables as recommended throughout the phases of the method. As such, all individuals who took part in the 4 Phases Method received standardized materials showing which foods should be avoided and which foods can be ingested during this study.

In association with food orientation, and to ensure adherence to the protocol, individuals are monitored weekly, for thirty minutes, by a trained and competent professional, a “health coach”, responsible for the food orientation. Weight loss is closely associated with understanding and adherence to dietary guidance provided by the “coach”, so that they correctly follow the 4 Phases Method. All health coaches are submitted to a standardized program of training, in which this professional is not only under supervision by the franchisor but also has medical support, in which all technical points (physiology, biochemistry, and epidemiology, among other aspects) and practical points (e.g., check anamnesis, step-by-step food orientation, and materials to be given) of the 4 Phases Method are presented. On completing all training steps, the health coach receives a certificate from the franchisor, which is valid for 2 years. After this period, all health coaches must revise their knowledge by repeating the training.

Lastly, the 4 Phases Method was developed and standardized so that it is applied in exactly the same way in all units of the EMAGRECENTRO network, a center that was founded more than 36 years ago in Brazil, focusing on health and well-being through procedures to promote weight loss among people with overweight and obesity.

### 2.2. Physical Activity

During Phase 1 and Phase 2 of the 4 Phases Methodology, it is recommended for all individuals to perform physical activity (with light to moderate intensity), preferentially aerobic activity (walking, 2–3 times per week for a duration of 15–30 min). In addition, all individuals performing Phases 3 and 4 receive standardized materials presenting options for physical activities, including 5 min of warmup, 30 min of aerobic exercises, and 5 min of stretching, to be performed 3–5 times per week. It is very important to highlight that, although not mandatory, physical activity is widely recommended.

### 2.3. Anthropometric Assessment Data

Anthropometric data [weight (kg), height (m), body mass index (BMI, kg/m^2^), and waist circumference (cm)] were assessed from the evaluation form on two different occasions: pre (baseline) and after completing the first 5 weeks of the 4 Phases Methodology (post).

### 2.4. Body Mass Index (BMI)

To assess the presence of people with underweight, normal weight, overweight, and obesity, the cutoff points for the body mass index (BMI = kg/m^2^), recommended both by the World Health Organization (WHO) and by the Ministry of Health of Brazil, were defined as BMI < 18.5 kg/m^2^ underweight; BMI 18.6–24.9 kg/m^2^ normal weight; BMI between 25 and 29.9 kg/m^2^ overweight; BMI > 30 kg/m^2^ obesity. BMI values between 30 and 34.9 kg/m^2^ are associated with grade I obesity, values between 35 and 39.9 kg/m^2^ with grade II obesity, and those above 40 kg/m^2^ with grade III obesity [1].

### 2.5. Urinary Ketone Bodies

Urinary ketone bodies were analyzed for presence or absence, a low-cost and a non-invasive method. A positive result is when urinary ketone bodies are detected in the urine and a negative result when urinary ketone bodies are not detected in the urine. According to the manufacturer’s instructions, for the urine dipstick to show a positive result, a concentration of >1.5 mM of urinary ketone bodies is necessary. Sodium nitroprusside and magnesium sulfate—which reacts with acetoacetate, a ketone body eliminated by urine—are present in the urine dipstick, and the dipstick changes from beige/light rose to violet/purple on presence of ketone bodies [18,19].

### 2.6. Statistical Analysis

Firstly, the normality of data was initially determined using the Shapiro–Wilk test and the homogeneity of variance was verified by Levene’s test.

Since the data showed a normal distribution, parametric variables (normal distribution) were presented as the mean and standard deviation (X_±_SD). In addition, in the intragroup analysis (baseline × post), we used the paired-T test; and in the intergroup analysis (women and men, or positive and negative, or between ages, or between BMI), we used repeated-measures ANOVA with Bonferroni’s post hoc test.

Effect size, used in the intergroup analysis, was calculated using Cohen’s coefficient with between 0.2 and 0.49 as a small effect; between 0.5 and 0.79 as a moderate effect; and more than 0.8 as a large effect [20].

In all analyses, the risk α ≤ 0.05 (*p* ≤ 0.05) was considered data analysis was performed using GraphPad Prism (version 8.1.2) software.

## 3. Results

Table 1 shows the anthropometric and physical characteristics of all participants in the present study, both grouped (total) and separated by sex (women and men), before (baseline) and after 5 weeks of application of the 4-Phases Methodology (post). Significant reductions in weight, BMI, and waist circumference were found post-application of the 4-Phases Methodology in total, women, and men. In addition, the weight and waist circumference values found in men were higher than observed in women both at baseline and post-application of the 4-Phases Methodology. All data obtained from each participant in this study were presented in the Appendix A.

Figure 2 shows the results concerning the weight (Figure 2A), BMI (Figure 2B), and waist circumference (Figure 2C) in all participants enrolled in the present study. In general, after the first 5 weeks of the 4 Phases Method application, significant reductions in these parameters were found (*p* < 0.0001, in Figure 2A, −7.68 ± 1.13 kg with ES = 0.32; in Figure 2B, −2.28 ± 0.33 BMI with ES = 0.36; in Figure 2C, −6.10 ± 2.69 cm with ES = 0.3). After 5 weeks of the 4 Phases Methodology, at least fifty percent of the subjects who took part in this methodology showed a decrease in one level of BMI (see Appendix A).

In order to verify whether gender influences responses to the 4 Phases Method, Figure 3 shows that both groups of women and men presented significant reductions in weight (*p* < 0.0001, −7.53 ± 1.02 kg with ES = 0.34 for women and −8.19 ± 1.65 kg with ES = 0.30 for men, Figure 3A), BMI (*p* < 0.0001, −2.84 ± 0.34 BMI with ES < 0.2 for women and −2.71 ± 0.28 BMI with ES < 0.2 for men, Figure 3B), and waist circumference (*p* < 0.0001, −7.67 ± 0.37 cm with ES = 0.39 for women and −7.40 ± 0.65 cm with ES = 0.41 for men, Figure 3C) after the methodology was applied. In addition, it was also found that not only was the weight of men higher than women both before (baseline, *p* < 0.001, ES = 0.56) and after (post, *p* < 0.001, ES = 0.56), but also the waist circumference of men was higher than women both before (baseline, *p* < 0.01, ES = 0.3) and after (post, *p* < 0.01, ES = 0.32), whereas the BMI was similar between the time points assessed.

Since age could also influence the parameters here evaluated, Figure 4 shows the results obtained in women and men separated by age (18ߝ40; 41ߝ60; >61 years). In terms of age, from 18 to 40 years, for both women and men, significant reductions in weight (*p* < 0.0001, −7.59 ± 1.31 kg with ES = 0.33 for women, Figure 4A; and *p* < 0.0001, −6.03 ± 2.68 kg with ES = 0.22 for men, Figure 4B), BMI (*p* < 0.0001, −2.81 ± 1.39 with ES = 0.33 for women, Figure 4C; and *p* < 0.0001, −2.81 ± 1.84 with ES = 0.43 for men, Figure 4D), and waist circumference (*p* < 0.0001, −8.04 ± 1.05 cm with ES = 0.42 for women, Figure 4E; and *p* = 0.0024, −7.4 ± 0.18 cm with ES = 0.43 for men, Figure 4F) have been observed post-4 Phases Method application. Similarly, for age 41 to 60 years, for both women and men, lower weight (*p* < 0.0001, −7.40 ± 1.04 kg with ES = 0.31 for women, Figure 4A; and *p* < 0.0001, −7.48 ± 2.15 kg with ES = 0.27 for men, Figure 4B), BMI (*p* < 0.0001, −2.78 ± 0.31 with ES = 0.37 for women, Figure 4C; and *p* < 0.0001, −2.6 ± 0.46 with ES = 0.40 for men, Figure 4D), and waist circumference (*p* = 0.0034, −7.53 ± 0.91 cm with ES = 0.36 for women, Figure 4E; and *p* = 0.022, −7.2 ± 1.48 cm with ES = 0.38 for men, Figure 4F) were found post-4 Phases Method application. For participants >60 years old, for women, a significant reduction in weight (*p* < 0.001, −7.26 ± 1.25 kg with ES = 0.27, Figure 4A), BMI (*p* < 0.001, −2.92 ± 0.49 with ES = 0.27, Figure 4C), and waist circumference (*p* = 0.049, −10.02 ± 1.44 cm with ES = 0.65, Figure 4C) was observed after methodology application. Although we found a toward reductions in weight and BMI in men, the statistical analysis did not present significant differences since the number of participants in this group decreased (*n* = 3).

Beyond the evaluation concerning age, we also performed an analysis of weight loss related to BMI (normal, overweight, obese I, obese II, and obese III, Figure 5). In general, there were significant reductions (**** *p* < 0.0001) in the weight for all BMI classes, with a small or moderate effect size, as shown in Figure 5A (total), Figure 5B (women), and Figure 5C (men). However, it was not possible to verify whether men with a normal BMI lost weight due to decreased number of participants (*n* = 2).

Lastly, we also analyzed the effect of the presence of urinary ketone bodies on the reduction in weight, BMI, and waist circumference (Figure 6). Interestingly, significant reductions in weight (*p* < 0.0001, −7.95 ± 1.63 kg with ES = 0.30, Figure 6A), BMI (*p* < 0.0001, −2.98 ± 0.90 BMI with ES = 0.37, Figure 6B), and waist circumference (*p* = 0.048, −6.87 ± 0.72 cm with ES = 0.33, Figure 6C) were observed post-4 Phases Method application in the volunteer groups that did not present urinary ketone bodies (negative). In a similar way, significant reductions in weight (*p* < 0.0001, −7.33 ± 0.57 kg with ES = 0.33, Figure 6A), BMI (*p* < 0.0001, −2.71 ± 0.15 BMI with ES = 0.37, Figure 6B), and waist circumference (*p* = 0.0015, −7.66 ± 0.67 cm with ES = 0.47, Figure 6C) were observed post-4 Phases Method application in the volunteer groups that presented urinary ketone bodies (positive).

## 4. Discussion

In the present study, our results showed that the 4 Phases methodology was able to significantly decrease all parameters analyzed, after 5 weeks of application, regardless of age, gender, and BMI. In addition, we can putatively suggest that the assistance of a health professional called a “health coach”, who was responsible not only for specific food orientation based on a ketogenic diet but also to stimulate participants to follow the methodology proposed by each one, could be a cornerstone to volunteers achieving their goals.

As a pillar of the 4 Phases Methodology, the ketogenic diet is broadly used in order to induce weight loss, particularly to promote health effects, such as reducing obesity, glycemia, insulin resistance, blood pressure, and triglycerides [21,22]. The use of diets that advocate a lower carbohydrate intake, such as the ketogenic diet, has grown in popularity in recent years since the failure of a low-fat diet to avoid an obesity epidemic was reported [7,23]. Interestingly, low-carbohydrate diets have been used to treat people with obesity for over 150 years—for instance, in the report published in 1863 by William Banting, which describes that an individual lost weight when on a low-carbohydrate diet [24]. In particular, the ketogenic diet is based on the replacement of sugar by ketone bodies to promote fuel energy without catabolism [25] due to specific dietary attributes, mainly the content of macronutrients (carbohydrates, fats, and proteins). Overall, the composition of KD is high in fat (50–60%, or 15–30 g fat/day), moderate in protein (30–35%, or 1–1.5 g protein per kg body weight) and low in carbohydrates (5–10%; in this case, usually less than 50 g/day) [7,16].

As mentioned, ketone bodies are produced when on the KD to guarantee enough energy to maintain proper function [26]. Therefore, measuring urinary ketone bodies is a low-cost and non-invasive strategy frequently used to verify whether individuals are following the KD [19]. Based on this fact, in the present study, we also evaluated urinary ketone bodies and it was observed that participants with negative results for the presence of urinary ketone bodies also presented a reduction in weight, BMI, and waist circumference as compared to participants who presented positive urinary ketone bodies. Although it is expected that participants with urinary ketone bodies could present higher reductions in the parameters here evaluated, it is noteworthy to point out that during the weight loss process, some people tend to autosabotage themselves by eating food that are not allowed in the 4 Phases Methodology, resulting in negative urinary ketone body tests, usually for 48 h even after being able to reduce weight, BMI and waist circumference, as observed in the present study. Furthermore, the absence of significant differences between the groups with or without urinary ketone bodies also could be attributed to the fact that this analysis mostly evaluated acetoacetate, one type of ketone body [27,28,29].

Beyond this observation, in terms of ketone bodies, another point that requires attention is associated to the fact that, regardless of BMI, gender, and age, a significant reduction in weight was found in all volunteers. In agreement with the literature, the KD, in the space of a short time (up to 4 weeks), was able to promote significant weight loss, particularly in overweight and obese individuals [7,9,15,16,21,24,30,31,32]. In a similar way, the studies that aimed to evaluate the effectiveness of long-term KD also demonstrated significant weight loss, especially in type 2 diabetes and the obese population [31]. Based on these pieces of information, in general, the remarkable effects of KD in inducing weight loss are closely related to the presence of diabetes and obesity, with little data concerning individuals with a normal BMI. Therefore, our results are important, since, as mentioned, individuals with a normal BMI can experience significant reductions in weight loss similar to people with overweight and obesity.

The capacity of the 4 phases method to decrease weight regardless of BMI was associated with significant reductions in waist circumference found in all individuals participating in this study. In fact, this last information corroborates our proposal that the 4 Phases methodology leads to significant weight loss in terms of body fat since the reductions in waist circumference indicate a decrease in abdominal fat [33,34].

As appealing as the results are, there is no guarantee that all individuals achieve real weight loss. According to the literature, factors that may impair individuals’ ability to achieve their goals are limited knowledge about correct food ingestion and how to implement food habits [35] as well as a lack of consistency throughout the whole weight loss process [36] and motivation [37] to follow the orientation, especially avoiding attitudes opposite to the diet recommendations proposed [37,38,39].

In order to promote success in weight loss associated with food orientation, it was reported that health coach assistance could represent the cornerstone to achieving real weight loss [39,40,41]. In terms of a health coach, according to Gershkowitz and collaborators (2020), this professional is responsible for “improving knowledge, goal setting, personalized feedback through frequent follow-up, and self-help techniques that allow for progress tracking” [40,42].

In the literature, the overwhelming majority of studies have demonstrated that health coach assistance leads to significant weight loss [42], although some authors mention that the effectiveness of health coach assistance in weight management is not fully clear [43].

Based on these pieces of information, in this study, the significant reduction in weight, BMI, and waist circumference found, regardless of age and gender, can be attributed to the presence of health coaching. In fact, health coach assistance is another pillar included in the 4 phases methodology, and is responsible for giving support to the customer; as formerly cited, these health care professionals are trained to understand the physiology, biochemistry, and methodology applied in the 4 Phase Method.

Another point that needs to be highlighted is related to the fact that some data used to perform this study were obtained before and during the COVID-19 pandemic. Particularly during the COVID-19 pandemic, health coach assistance was performed remotely via (telehealth), whereas assistance was exclusively performed in person before. There is no doubt that the use of different digital platforms is useful to guarantee health coach assistance since it was demonstrated that health coach assistance can promote good outcomes and significant weight loss, in a similar way to that observed with in-person assistance [44,45,46].

## 5. Limitations of This Study

The limitations of the present study were: (a) in order to develop the present study, we initially obtained data from more than 1200 people who were took part to the 4 Phases Methodology, but, unfortunately, only data from 354 people contained all the information required to perform this study; (b) the reduced number of older people (>60 years old), especially men (*n* = 3), which did not allow us to verify whether the weight, BMI, and waist circumference were significantly altered on 4 Phases Methodology application; and (c) comparing results for with and without health coach assistance (a control group) could be useful to prove the actual effectiveness of this professional in achieving a significant reduction in body weight, BMI, and waist circumference.

## 6. Conclusions

In conclusion, our results suggest that the scientific 4 Phases Methodology is effective in promoting a significant reduction in body weight (mean: −8.8%/−7.6 kg), BMI, and waist circumference(mean: −7.1%/−7.4 cm), in the short term, regardless of age, gender, and BMI, with an association between a ketogenic intake strategy with regular weekly health coach assistance. In addition, the 4 Phases Method represents an important non-pharmacological intervention to revert and/or mitigate the development and progression of obesity with safety, as well as its harmful consequences in all populations.

## 7. Patents

Scientific Weight Loss—4 Phases Method is registered at the U.S. Copyright Office under the registration numbers 2-056-931 and 2-071-992. It is also registered in the Brazilian National Library under registration number 764269, book 1482, and page 311.

## Figures and Tables

**Figure 1 healthcare-10-01398-f001:**
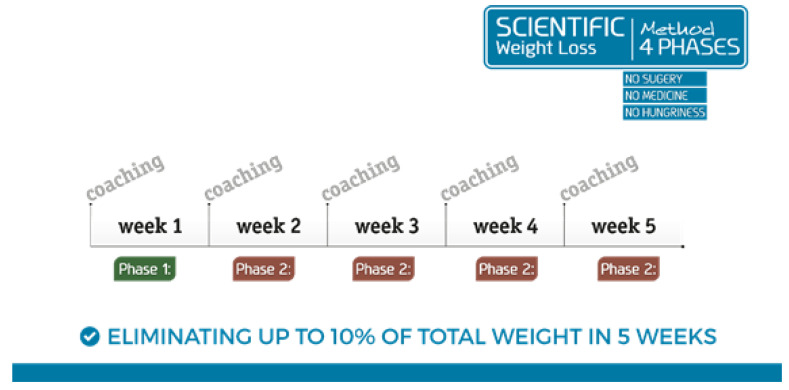
Representative design of the 4 Phases Method applied in this study.

**Figure 2 healthcare-10-01398-f002:**
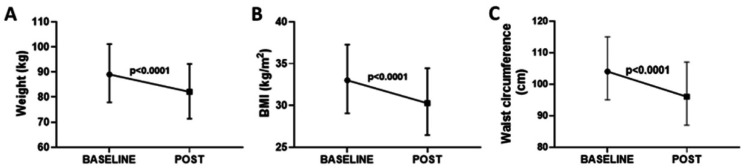
Representation of variation in weight (kg, (**A**)), BMI (kg/m^2^, (**B**)), and waist circumference (cm, (**C**)), presented as the mean and SD, of all the volunteers enrolled in the present study, both before (BASELINE) and after (POST) 4 Phases Methodology application.

**Figure 3 healthcare-10-01398-f003:**
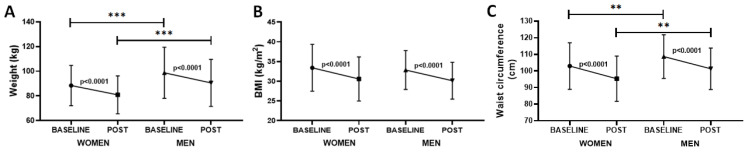
Representation of variation in weight (kg, (**A**)), BMI (kg/m^2^, (**B**)), and waist circumference (cm, (**C**)), presented as the mean and SD, of the volunteers enrolled in the present study separated into women and men, both before (BASELINE) and after (POST) 4 Phases Methodology application. ** *p* < 0.01 *** *p* < 0.001.

**Figure 4 healthcare-10-01398-f004:**
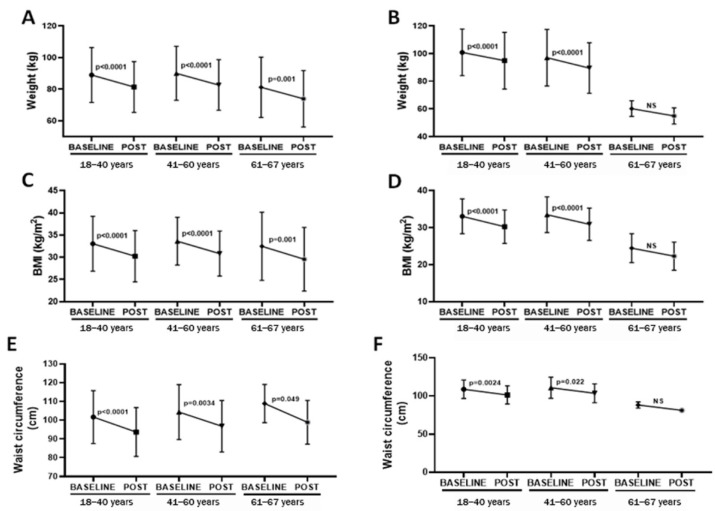
Representation of variation in weight (kg, (**A**,**B**)), BMI (kg/m^2^, (**C**,**D**)), and waist circumference (cm, (**E**,**F**)), presented as the mean and SD, of the volunteers enrolled in the present study separated initially into women (**A**,**C**,**E**) and men (**B**,**D**,**F**), followed by separation into different ages (from 18 to 40; from 41 to 60; and more than 60 years), both before (BASELINE) and after (POST) 4 Phases Methodology application.

**Figure 5 healthcare-10-01398-f005:**
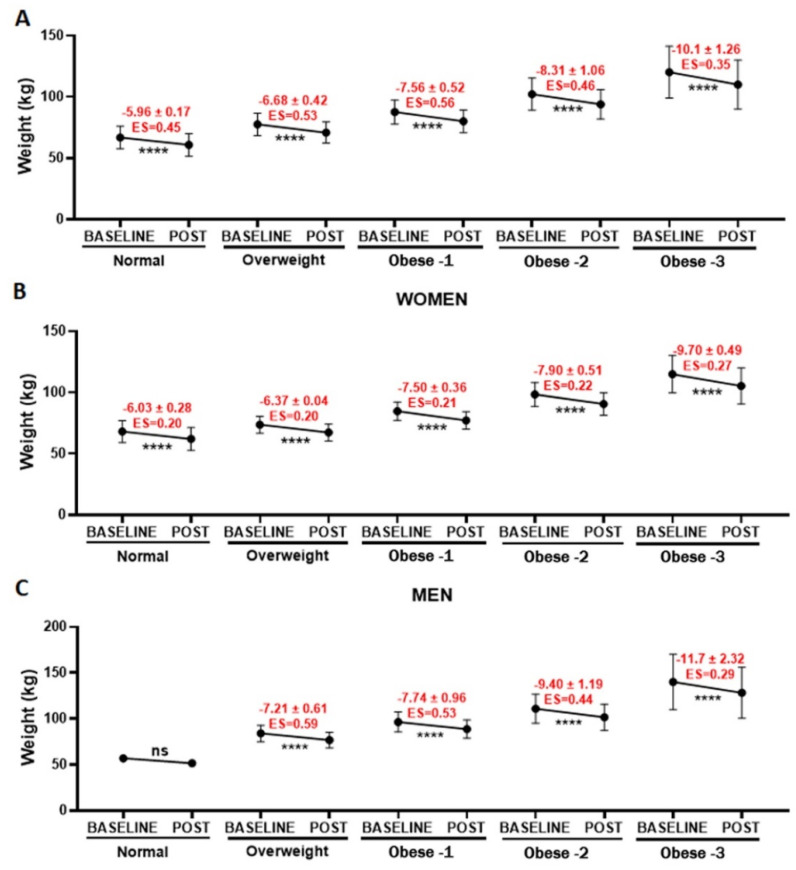
Representation of variation in weight (kg), presented as the mean and SD, of all the volunteers enrolled in the present study (**A**) or separated into women (**B**) and men (**C**), followed by separation by different BMI (normal—18.6–24.9 kg/m^2^; overweight—25 to 29.9 kg/m^2^; obese I—>30 kg/m^2^; obese II—35 and 39.9 kg/m^2^; and obese III—40 kg/m^2^), both before (BASELINE) and after (POST) 4 Phases Methodology application. **** *p* < 0.0001.

**Figure 6 healthcare-10-01398-f006:**
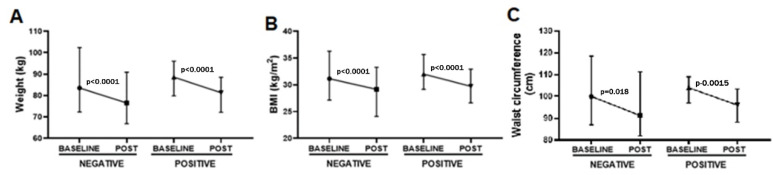
Representation of variation in weight (kg, (**A**)), BMI (kg/m^2^, (**B**)), and waist circumference (cm, (**C**)), presented as the mean and SD, of all the volunteers enrolled in the present study separated into two different groups according to the presence (POSITIVE) or absence (NEGATIVE) of urinary ketone bodies, both before (BASELINE) and after (POST) 4 Phases Methodology application.

**Table 1 healthcare-10-01398-t001:** Anthropometric (age) and physical characteristics (height, weight, BMI, and waist circumference), presented as the mean and standard deviation (X_±_SD) of total participants and separated by sex (women and men) before (baseline) and after 5 weeks of application of the 4 Phases Methodology (post).

Variables	Total (*n* = 354)		*p*-Value	Women (*n* = 260)		*p*-Value	Men (*n* = 94)		*p*-Value	*p*-Value	*p*-Value
	Baseline	Post		Baseline	Post		Baseline	Post		W versus M Baseline	W versus M Post
Age (years)	38.5 ± 10.1	-	NS	38.8 ± 11	-	NS	37.5 ± 10.6	-	NS	NS	NS
Height (m)	1.65 ± 0.09	1.65 ± 0.09	NS	1.63 ± 0.08	1.63 ± 0.08	NS	1.72 ± 0.09	1.72 ± 0.09	NS	NS	NS
Weight (kg)	92.2 ± 19.8	84.4 ± 18.6	0.0001	88.2 ± 17.9 *	81.6 ± 16.8 *	0.0001	99.7 ± 22.4	91.5 ± 20.6	0.0001	0.0001	0.0003
BMI (kg/m^2^)	33.4 ± 6.1	30.6 ± 5.7	0.0001	33.3 ± 6.2	30.5 ± 5.9	0.0001	33.5 ± 6.5	30.7 ± 6.1	0.0001	NS	NS
Waist circumference (cm)	104.8 ± 14.3	97.2 ± 13.9	0.0001	103 ± 14 *	95.3 ± 13.6 *	0.0001	109.2 ± 14.1	101.9 ± 13.6	0.0001	0.0027	0.0014

* Significant difference in relation to the values found in men.

## Data Availability

Data availability is under the responsibility of the authors and data may be made available upon request and need.

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
