# Peer review of "From Young to Older, the 4 Phases Method Is Efficient in Promoting Quick Weight, BMI, and Waist Circumference Reductions"

_healthcare, 2022, doi:10.3390/healthcare10081398_

Round 1

Reviewer 1 Report

Overall comment: I strongly recommend a language check (especially regarding word order and word choice) to enhance fluency in all sections of the paper. 

Introduction: The introduction gives an overview on overweight and obesity and strategies (ketogenic diet) to promote weight loss. However, the effect of "health coach" is not regarded in the introduction, but stated as main intervention effect in the discussion and conclusion section. 

L 83 to 85 and 89 to 91: I don't think it is necessary to state such precise results  in the introduction section. 

Methods: it is stated that EMAGRECENTRO is a clinic specialized in weight loss and aesthetic procedures. Hence, the retrospective data obtained is patients data. Within this section, this seems to be neglected. Could you please explain why? Furthermore, it is not stated who initially analyzed the first data collection of the 119 franchises. Who was responsible for selecting the data? I would assume a physician/medical expert? Please state. 

concerns about ethical vote: since these are medical records obtain from a clinic, please justify, why a vote from a medical ethics committee was not obtained. 

Line 123-124: who are "the responsible for the EMAGRECENTRO's franchises"? Please state.

Line 148 who gives this medical advice?

Line 163: could you please state a reference here?

Line 182 ff: it is not clear if the health coach completed the standardized training program before coaching the participants or if he/she was still in training

Line 197: did you track physical activity behaviour? If yes, could you please state how? 

Results: Table 1: please delete the repetition in the sections "age", there is no post value.

Figure 3 presents the same results than stated in table 1 concerning weight, BMI and waist ciricumference. This is a double display of results. I recommend eliminating main outcomes concerning weight, BMI and waist cirucmference from table 1. 

Discussion: L355-358: please be careful using this statement. Since you did not include a control group (a group without health coach), the effect of the coaches work can not be evaluated scientifically. (also regard L431 and the conclusion section)

L360-372: this section includes a description of ketogenic diet rather belonging to the introduction section. Could you please state the relevance of this section to your results? 

Author Response

 Overall comment: I strongly recommend a language check (especially regarding word order and word choice) to enhance fluency in all sections of the paper. 

Authors´ comment: First of all, we would like to thank the reviewer for the revision and also for the criticisms and suggestions that allowed us to improve our manuscript. In addition, we would like to inform you that all the responses to your questions raised are presented below and also that the revised sentences/paragraphs are marked in yellow in the main text.

----------------------------------------------------------------------------------------

Introduction: The introduction gives an overview on overweight and obesity and strategies (ketogenic diet) to promote weight loss. However, the effect of "health coach" is not regarded in the introduction, but stated as main intervention effect in the discussion and conclusion section. 

Authors´ response: We would like to thank you for the comment and to clarify that the "Introduction" section was revised and, as recommended, two new sentences about “health coach” effects were added in this section, as presented below.

"Furthermore, the authors also highlighted that in this intervention, the presence of a health professional, as a health coach, is a safe approach to avoid common side effects that are clinically mild and easy to control, in which the recovery is often spontaneous. Besides, the health coach is also a fundamental piece of behavior modification that seems more effective in treating obesity than any traditional method, but it has not been widely applied.”

----------------------------------------------------------------------------------------

L 83 to 85 and 89 to 91: I don't think it is necessary to state such precise results in the introduction section. 

Authors´ response: We would like to thank you for your comment and clarify that, in line with a previous round of manuscript review, it was recommended to add some information about effect estimates for each of these outcomes (to demonstrate the significance of effect and duration and what kind of other interventions of weight loss were). Therefore, we choose not to remove these pieces of information.

----------------------------------------------------------------------------------------

Methods: it is stated that EMAGRECENTRO is a clinic specialized in weight loss and aesthetic procedures. Hence, the retrospective data obtained is patients data. Within this section, this seems to be neglected. Could you please explain why? Furthermore, it is not stated who initially analyzed the first data collection of the 119 franchises. Who was responsible for selecting the data? I would assume a physician/medical expert? Please state. 

Authors´ response: We would like to thank you for the comment and to clarify that, as reported, to perform this retrospective study we first obtained the Ethics Committee's approval. After that, we contacted the responsible for the EMAGRECENTRO´s franchises not only to explain the study but also to ask for help to obtain the data necessary for the study´s development. In this contact, we also asked the responsible for the EMAGRECENTRO´s franchises to provide us the written informed consent from customers who wanted to participate in the study, thus allowing us to use their medical records. In relation to those customers who weren´t able to be contacted by the responsible for the EMAGRECENTRO´s franchises (due to changed address, e-mail, and cellphone/telephone number), we obtained the written informed consent for use of those data directly from the responsible for the EMAGRECENTRO´s franchises through a specific document, which is a legal and mandatory way for research´s development that will collect information in a database of institutions in Brazil. Moreover, it is important to reinforce that all the data provided by the responsible for EMAGRECENTRO´s franchises were used after we obtained the written informed consent, and no personal information was sought. Lastly, written informed consents were approved by the Ethics Research Committee of Santo Amaro University (number:4.928.459) and sent to MDPI. The responsibility for selecting the data was one of the authors, which is a physician and a certified ketogenic nutrition specialist.

----------------------------------------------------------------------------------------

concerns about ethical vote: since these are medical records obtain from a clinic, please justify, why a vote from a medical ethics committee was not obtained. 

Authors´ response: We would like to thank you for the comment and also to mention that all data of the participants were used exclusively after signing the informed consent by the customers or by the responsible for the EMAGRECENTRO´s franchises (which is a legal and mandatory way for research´s development that will collect information in a database of institutions in Brazil). In order to perform this study, only after the Ethics Committee gave us the approval, we contacted the responsible for these clinics and asked for the data contained in the medical records for the development of the study.

----------------------------------------------------------------------------------------

Line 123-124: who are "the responsible for the EMAGRECENTRO's franchises"? Please state.

Authors´ response: We would like to thank you for the comment and also to clarify that, as reported in the previous response, to perform the present study we only used data from 354 individuals, named as individuals or participants in the main text, which were provided by the responsible for the EMAGRECENTRO´s franchises. In addition, it is noteworthy to mention that all data were in the medical records of the customers and were used exclusively after signing the informed consent by the customers or by the responsible for the EMAGRECENTRO´s franchises.

Regarding the question raised about EMAGRECENTRO´s franchises, we would like to inform you that all of them are health and aesthetic centers and in agreement with Brazilian laws is not mandatory the presence of medical professionals in these centers. However, in all franchisees, there is a presence of professionals related to the health area, such as biomedical.

----------------------------------------------------------------------------------------

Line 148 who gives this medical advice?

Authors´ response: We would like to thank you for the comment and to clarify that in accordance with the information provided us with the EMAGRECENTRO´s franchises, all the health coaches are in contact with medical team support if necessary. The health coach is not there to give any medical advice, but they are there to give limited orientation about the 4 Phases Methodology. All health coaches submitted to the standardized training, which is divided into two phases: one in-person and the other remotely. In both phases, there is not only the supervision of the franchisor as well as medical support, in which all technical points (physiology, biochemistry, epidemiology, among other aspects) of the 4 Phases Method are explained. After the individual in training obtains theoretical information concerning the 4 Phases Method, this professional is submitted to an observational internship in the EMAGRECENTRO´s franchises, where he or she can follow a certified health coach in order to observe and learn some practical points associated with the application of this methodology. Lastly, this professional in training is oriented to do an online course about not only the 4 Phases Methodology but also the health coach obligations. After completing all these steps, this recent health coach is able to receive a certificate from the franchisor, and its certificate is valid for 2 years. After this period, all health coaches are needed to recycle their knowledge by repeating the training.

----------------------------------------------------------------------------------------

Line 163: could you please state a reference here?

Authors´ response: We would like to thank you for the comment and, as suggested, a reference was added.

----------------------------------------------------------------------------------------

Line 182 ff: it is not clear if the health coach completed the standardized training program before coaching the participants or if he/she was still in training

Authors´ response: We would like to thank you for the comment and to clarify that paragraph was revised and corrected, as presented below.

“Since completing all training steps, the health coach is able to receive a certificate from the franchisor, and its certificate is valid for 2 years, with that the health coach is able to be responsible for the food orientation of 4 Phases Method”

----------------------------------------------------------------------------------------

Line 197: did you track physical activity behaviour? If yes, could you please state how? 

Authors´ response: We would like to thank you for the comment and, in response to the question raised, to clarify that, in agreement with the information provided by the responsible for the EMAGRECENTRO´s franchises, physical activity behavior is tracked by a standardized material during the consultation, to register whether the individuals did or did not perform physical activity the week before. As mentioned in the main text, the participants scheduled once a week an appointment with the 4 Phases Method Heath Coach to analyze this parameter, too.

----------------------------------------------------------------------------------------

ResultsTable 1: please delete the repetition in the sections "age", there is no post value.

Authors´ response: We would like to thank you for the comment and, as recommended, we deleted the repetition in the section “age” in the "table 1" in the "Results section", as presented below.

Variables

Total (n=354)

p-value

Women (n=260)

p-value

Men (n=94)

p-value

p-value

p-value

Baseline

Post

Baseline

Post

Baseline

Post

WxM

Baseline

WxM

Post

Age (years)

38.5±10.1

-------

NS

38.8±11

-------

NS

37.5±10.6

-----------

NS

NS

NS

Height (m)

1.65±0.09

1.65±0.09

NS

1.63±0.08

1.63±0.08

NS

1.72±0.09

1.72±0.09

NS

NS

NS

Weight (kg)

92.2±19,8

84.4±18.6

0.0001

88.2±17.9*

81.6±16.8*

0.0001

99.7±22.4

91.5±20.6

0.0001

0.0001

0.0003

BMI (kg/m2)

33.4±6.1

30.6±5.7

0.0001

33.3±6.2

30.5±5.9

0.0001

33.5±6.5

30.7±6.1

0.0001

NS

NS

Waist circumference (cm)

104.8±14.3

97.2±13.9

0.0001

103±14*

95.3±13.6*

0.0001

109.2±14.1

101.9±13.6

0.0001

0.0027

0.0014

----------------------------------------------------------------------------------------

Figure 3 presents the same results than stated in table 1 concerning weight, BMI and waist ciricumference. This is a double display of results. I recommend eliminating main outcomes concerning weight, BMI and waist cirucmference from table 1. 

Authors´ response: We would like to thank you for the comment and to inform you that the Table 1 was revised, as formerly recommended, and also to clarify that the presentation of the main data obtained in general both in Table 1 and Figure 3 was based on the previous suggestion provided by reviewers in order to highlight the values and representative description of them. Thus, we opt to maintain this double presentation to offer a good way to evidence our results.

----------------------------------------------------------------------------------------

Discussion: L355-358: please be careful using this statement. Since you did not include a control group (a group without health coach), the effect of the coaches work can not be evaluated scientifically. (also regard L431 and the conclusion section)

Authors´ response: We would like to thank you for the comment and, as recommended, the following paragraphs were revised, as presented below.

In addition, we can putatively suggest that the assistance of a health professional called "coach", which was responsible not only for specific food orientation based on a ketogenic diet but also to stimulate the participants to follow the methodology proposed by each one, could be a cornerstone to the volunteers to achieve their goals.”

“In conclusion, our results allowed us to suggest that the scientific 4 Phases Methodology is effective in promoting a significant reduction in body weight, BMI, and waist circumference in the short term, regardless of age, gender, and BMI classification, due to this methodology to be based on an association between a ketogenic intake strategy with regular weekly assistance by a health coach.

----------------------------------------------------------------------------------------

L360-372: this section includes a description of ketogenic diet rather belonging to the introduction section. Could you please state the relevance of this section to your results? 

Authors´ response: We would like to thank you for the comment and also to clarify that in the first paragraph of "Discussion" section, we opt to describe the main results obtained in the present study in order to in a sequence to provide pieces of information that can corroborate or not our findings. In addition, in the first sentence in the second paragraph, we presented more information concerning the relationship between the 4 Phases Methodology and the pillar of this method, which is the ketogenic diet. Thus, these pieces of information are crucial to the development of the "Discussion" section and were not equal to present in the "Introduction" section, even though were related to the same context.

Reviewer 2 Report

The authors aimed to  investigate the effectiveness of a methodology ( they call a Scientific 4 Phases Method, developed by EMAGRECENTRO, which is based on a ketogenic approach (total carbohydrate intake <40g/day; including fibers), in conjunct with health coach assistance, to promote a reduction in body weight, body mass index (BMI), and waist circumference after 5 weeks pro-gram duration.

They recruited  354 individuals, both sexes, aged between 18-67.

Age, sex, weight, height, BMI, waist circumference measurement, and the presence of ketone bodies in the urine were evaluated before (baseline) and after 5 weeks of the 4 Phases application

They concluded that their results demonstrated that the 4 Phases Method was able to promote significant weight, BMI, and waist circumference reductions in the short-term, mainly by associating a ketogenic intake strategy with a regular close follow-up weekly consultation with a health coach assistance

The article faces an interesting topic and flows well.

I have some minor comments

1.      Some passages are hard to follow (starting from the first sentence in the abstract). Please smooth them

2.      Numbers are important. However, please consider to improve the way to presenting them. See for example the text “  Beyond the evaluation concerning age, it was also performed the analysis of weight 315

loss related to the BMI stratification (Normal, Overweight, Obese I, Obese II, and Obese 316

III, Figure 5). As shown in Figure 5A (total), Figure 5B (women group), and Figure 5C 317

(men group), in a general way there were significant reductions (p<0.0001) in the weight 318

for all BMI stratification, with small or moderate effect size (in Figure A – Normal, - 319

5.96±0.17 kg with ES=0.45; Overweight, -6.68±0.42 kg with ES=0.53; Obese I, -7.56±0.52 kg 320

with ES=0.56; Obese II, -8.31±1.06 kg with ES=0.46; Obese III, -10.1±1.26 kg with ES=0.35; 321

in Figure B – Normal, -6.03±0.28 kg with ES=0.20; Overweight, -6.37±0.04 kg with ES=0.20; 322

Obese I, -7.50±0.36 kg with ES=0.21; Obese II, -7.90±0.51 kg with ES=0.22; Obese III, - 323

9.70±0.49 kg with ES=0.27; and in Figure C – Overweight, -7.21±0.61 kg with ES=0.59; 324

Obese I, -7.74±0.96 kg with ES=0.53; Obese II, -9.40±1.19 kg with ES=0.44; Obese III, - 325

11.7±2.32 kg with ES=0.29). It is worth highlighting that it was not possible to verify 326

whether the men with normal BMI lost weight due to decreased number of participants 327

(n=2).”

3.      Please join the “2. Methods” with the “3.phase methods”

4.      Insert the limittaions

5.      Perhaps you could insert one of two sentences on the perspectives in the conclusions

Author Response

The authors aimed to  investigate the effectiveness of a methodology ( they call a Scientific 4 Phases Method, developed by EMAGRECENTRO, which is based on a ketogenic approach (total carbohydrate intake <40g/day; including fibers), in conjunct with health coach assistance, to promote a reduction in body weight, body mass index (BMI), and waist circumference after 5 weeks program duration.

They recruited 354 individuals, both sexes, aged between 18-67.

Age, sex, weight, height, BMI, waist circumference measurement, and the presence of ketone bodies in the urine were evaluated before (baseline) and after 5 weeks of the 4 Phases application

They concluded that their results demonstrated that the 4 Phases Method was able to promote significant weight, BMI, and waist circumference reductions in the short-term, mainly by associating a ketogenic intake strategy with a regular close follow-up weekly consultation with a health coach assistance

The article faces an interesting topic and flows well.

I have some minor comments

Authors´ comment: First of all, we would like to thank the reviewer for the revision and also for the criticisms and suggestions that allowed us to improve our manuscript. In addition, we would like to inform you that all the responses to your questions raised are presented below and also that the revised sentences/paragraphs are marked in blue in the main text.

----------------------------------------------------------------------------------------

  1. Some passages are hard to follow (starting from the first sentence in the abstract). Please smooth them

Authors´ comment: We would like to thank you for the comment and to inform you that the description abstract was revised, as presented below.

Investigate the effectiveness of the Scientific 4 Phases Method, a methodology developed by EMAGRECENTRO, which is based on a ketogenic approach (total carbohydrate intake <40g/day; including fibers) associated with health coach assistance, in promoting reduction in body weight, body mass index (BMI), and waist circumference after 5 weeks of methodology application.”

----------------------------------------------------------------------------------------

  1. Numbers are important. However, please consider to improve the way to presenting them. See for example the text “  Beyond the evaluation concerning age, it was also performed the analysis of weight 315

loss related to the BMI stratification (Normal, Overweight, Obese I, Obese II, and Obese 316

III, Figure 5). As shown in Figure 5A (total), Figure 5B (women group), and Figure 5C 317

(men group), in a general way there were significant reductions (p<0.0001) in the weight 318

for all BMI stratification, with small or moderate effect size (in Figure A – Normal, - 319

5.96±0.17 kg with ES=0.45; Overweight, -6.68±0.42 kg with ES=0.53; Obese I, -7.56±0.52 kg 320

with ES=0.56; Obese II, -8.31±1.06 kg with ES=0.46; Obese III, -10.1±1.26 kg with ES=0.35; 321

in Figure B – Normal, -6.03±0.28 kg with ES=0.20; Overweight, -6.37±0.04 kg with ES=0.20; 322

Obese I, -7.50±0.36 kg with ES=0.21; Obese II, -7.90±0.51 kg with ES=0.22; Obese III, - 323

9.70±0.49 kg with ES=0.27; and in Figure C – Overweight, -7.21±0.61 kg with ES=0.59; 324

Obese I, -7.74±0.96 kg with ES=0.53; Obese II, -9.40±1.19 kg with ES=0.44; Obese III, - 325

11.7±2.32 kg with ES=0.29). It is worth highlighting that it was not possible to verify 326

whether the men with normal BMI lost weight due to decreased number of participants 327

(n=2).”

Authors´ comment: We would like to thank you for the comment and, as recommended, we revised the presentation of these data. In this regard, we would like to inform you that we inserted all the values into the Figure 5 and also altered the describing these results, as presented below:

"Beyond the evaluation concerning age, it was also performed the analysis of weight loss related to the BMI stratification (Normal, Overweight, Obese I, Obese II, and Obese III, Figure 5). In a general way, there were significant reductions (****p<0.0001) in the weight for all BMI stratification, with small or moderate effect size, as shown in Figure 5A (total), Figure 5B (women group), and Figure 5C (men group). It is worth highlighting that it was not possible to verify whether the men with normal BMI lost weight due to decreased number of participants (n=2)."

----------------------------------------------------------------------------------------

  1. Please join the “2. Methods” with the “3.phase methods”

Authors´ comment: We would like to thank you for the comment and, as suggested, we join both subtitles.

----------------------------------------------------------------------------------------

  1. Insert the limittaions

Authors´ comment: We would like to thank you for the comment and, as recommended, we inserted in the main text a new topic "5", named “Limitations of the study”, after the “Discussion” section, as presented below.

  1. Limitations of the study

The limitations of the present study were: a) in order to develop the present study, we initially obtained data from more than 1.200 people that were submitted to the 4 Phases Methodology, but, unfortunately, only data from 354 people contained all the information required to perform this study; b) the reduced number of older people (>60 years old), especially men group (n=3), which did not allow us to verify whether the weight, BMI, and waist circumference were significantly altered during the period of them were submitted to the 4 Phases Methodology; and c) the possibility to compare our results with a group of individuals that were also submitted to the 4 Phases Method-ology but without the assistance by the health coach (control group), could be useful to prove the actual effectiveness of this professional and its assistance in helping the significant loss of body weight, BMI, and waist circumference.”

----------------------------------------------------------------------------------------

  1. Perhaps you could insert one of two sentences on the perspectives in the conclusions

Authors´ comment: We would like to thank you for the comment and, as suggested, we added a new sentence concerning the perspectives, as presented below.

“In addition, in view of these prominent results, we can also point out that the 4 Phases Method could represent an important non-pharmacological intervention to revert and/or mitigate the development and progression of obesity, as well as its harmful consequences in all population."

Round 2

Reviewer 1 Report

Thank you for your thouroughly replies. You gave me a better understanding of the system in Brazil in which an institution like that operates. Therefore, my major ethical concerns are gone. 

You revised the comments thouroughly and supplied valid arguments if you chose not to change sections recommended.